# Identification of New Toxicity Mechanisms in Drug-Induced Liver Injury through Systems Pharmacology

**DOI:** 10.3390/genes13071292

**Published:** 2022-07-21

**Authors:** Aurelio A. Moya-García, Andrés González-Jiménez, Fernando Moreno, Camilla Stephens, María Isabel Lucena, Juan A. G. Ranea

**Affiliations:** 1Departamento de Biología Molecular y Bioquímica, Universidad de Málaga, 29071 Málaga, Spain; ranea@uma.es; 2Laboratorio de Biología Molecular del Cáncer, Centro de Investigaciones Médico-Sanitarias Universidad de Málaga, 29071 Málaga, Spain; 3Instituto de Investigación Biomédica de Málaga (IBIMA), 29590 Málaga, Spain; kolidri@gmail.com (A.G.-J.); cstephens@uma.es (C.S.); lucena@uma.es (M.I.L.); 4Servicio de Farmacología Clínica, Hospital Universitario Virgen de la Victoria, Facultad de Medicina, Universidad de Málaga, 29010 Málaga, Spain; 5Departamento de Arquitectura de Computadores, Universidad de Málaga, 29071 Málaga, Spain; fmjabato@gmail.com; 6LifeWatch ERIC Spain, 41071 Sevilla, Spain; 7Servicio de Supercomputacion, Universidad de Málaga, 29071 Málaga, Spain; 8Centro de Investigación Biomedica en Red de Enfermedades Hepaticas y Digestivas (CIBERehd), 28029 Madrid, Spain; 9Centro de Investigación Biomedica en Red de Enfermedades Raras (CIBERER), 29029 Madrid, Spain; 10Spanish National Bioinformatics Institute (INB/ELIXIR-ES), 08034 Barcelona, Spain

**Keywords:** drug-induced liver injury, toxicology, polypharmacology, systems pharmacology

## Abstract

Among adverse drug reactions, drug-induced liver injury presents particular challenges because of its complexity, and the underlying mechanisms are still not completely characterized. Our knowledge of the topic is limited and based on the assumption that a drug acts on one molecular target. We have leveraged drug polypharmacology, i.e., the ability of a drug to bind multiple targets and thus perturb several biological processes, to develop a systems pharmacology platform that integrates all drug–target interactions. Our analysis sheds light on the molecular mechanisms of drugs involved in drug-induced liver injury and provides new hypotheses to study this phenomenon.

## 1. Introduction

Adverse drug reactions pose a significant burden on hospital admissions, hospitalisation time and related emergencies [1]; they are also a major cause of market withdrawals [2]. Drug-induced liver injury (DILI) presents a particular challenge, for it has diverse, complex and largely uncharacterised mechanisms [3], and it is challenging to diagnose and assess in the population [4].

Our current knowledge of DILI is limited to research areas such as the analysis of cases registries [4]; biomarkers of liver toxicity—including small-hairpin RNAs [5]; QSAR toxicity models [6]; genetic risk factors [7,8]—along with null phenotypes for detoxification enzymes [9] or reduced levels of the hepatocyte bile salt export pump linked to the accumulation of bile acids in the liver [10]; and alterations in drug metabolism [11]. Besides, DILI research leans on the mechanistic study of individual drugs rather than analysing the complex interaction between drugs—for instance, research focused on amocillin-clavulanate [8], diclofenac [12], pazopanib [13] or isozianid [11]. Nevertheless, initiatives such as the DILI-sim Initiative aims to leverage bottom-up systems pharmacology, i.e., mechanistic, mathematical models of hepatotoxicity, to understand and predict DILI in humans and preclinical species, thus describing the effect of drug exposure on the hepatocyte, liver and whole organism [14].

Over the past decades, there has been a decrease in the translation of drug candidates into effective therapies, concurrent with the dominance of the assumption that a drug is a selective ligand on a single target—the so-called ‘magic bullets’ [15]. Therefore, the conjecture that a drug acts by targeting one particular and critical point or step in a disease process, thus affecting a cure with few other consequences, is an over-simplification that hampers our ability to comprehend how a drug operates and causes adverse reactions [16]. A growing body of evidence collectively grouped under the umbrella term ‘polypharmacology’, i.e., the idea that there are many drugs for each target and a single drug can affect multiple targets [17], challenges this paradigm, shedding light on the complexity of drug action. In the paradigm of network pharmacology, a drug can bind multiple targets with low specificity and therefore can perturb several biological processes playing numerous roles. In other words, it can have various modes of action (MoA), which are accountable for its polypharmacology. Therefore, a drug might be toxic because of its interactions with known therapeutic targets—for instance, because it is present in different body tissues. In addition, its interaction with other proteins could explain drug toxicity, i.e., a drug may be toxic because it is not selective enough, thus causing unexpected off-target effects. The analysis of this relationship between drug side-effects and their multiple targets can predict new drug targets through side-effect similarity [18], identify proteins that cause drug side-effects [19] and reveal the network properties of drug targets [20]. Furthermore, the concepts of polypharmacology and network pharmacology become settled in the idea that drug toxicity is a consequence of the biological systems the drug interacts with [21].

In a previous study, we analysed drug multi-target profiles to show that protein domains (i.e., the compact functional and structural units of proteins) might be the druggable elements within proteins. We proved that a drug could act on many different protein effectors because it targets a shared domain on all of them [17]. Thus, in this study, we develop a top-down systems pharmacology strategy that integrates domain-based and network-based approaches. We look into the ability of drugs to target different proteins to investigate the molecular mechanisms of liver toxicity. We modelled a drug network based on drug interactions with proteins and protein domains where drugs associate according to their mechanism of action. Further analysis of the drug network suggests that alterations in liver regeneration, liver metabolism and the immune response cause liver damage associated with drug intake, thus providing hypotheses that can guide further research into the molecular mechanisms that underlie DILI.

## 2. Materials and Methods

We base our analysis of the mechanisms involved in DILI on a drug network in which drugs link together if they share molecular targets. We consider the molecular targets of a drug, its intended pharmacological target and its off-targets. Therefore, we will assess DILI drugs’ mechanisms and other pathways and biological processes DILI drugs might participate in through their interaction with proteins different from their intended targets.

The annotation of the proteins that interact with drugs (targets and off-targets) is limited and biased towards a few well-characterised therapeutical targets; small molecules are not systematically screened through large panels of proteins because of limited time and resources [22]. Thus, targets annotated on public sources are more representative of the target space explored by the pharmaceutical industry rather than being a true reflection of all the targets a drug might have. Since protein domains mediate the interactions between drugs and their protein targets, they constitute a reasonable annotation level of how drugs interact with their targets [17]. Therefore, to make up for the limitation on the target space of DILI drugs, we have considered the Pfam domains [23] of those proteins as an additional source of drug target information.

To analyse the molecular mechanisms and biological processes in which DILI drugs are involved, we have (i) obtained and curated a set of drugs that elicit liver toxicity at different degrees; (ii) obtained the protein targets and the drug–protein interactions (dP) for these drugs from public sources; (iii) obtained the multi-domain architecture of these proteins from Pfam; (iv) inferred the drug–domain associations (dD) that stem from the dP above and the domain composition of the protein targets; (v) integrated the dP and dD data into a coherent drug-target dataset and (vi) modelled the drug network and analysed the communities of tightly connected drugs within the network.

### 2.1. Gathering Drug, Proteins and Domain Data

We compiled a list of 229 hepatotoxic drugs from the Spanish DILI Registry [24] using their summary products characteristic (SPC). Drugs were annotated based on their liver toxicity into very toxic drugs (drugs withdrawn from the market due to liver toxicity; 54 drugs), toxic drugs (drugs related to acute liver failure or with high rates of hepatotoxicity; 175 drugs) and non-toxic drugs (drugs with low risk or low rates of hepatotoxicity). In addition, we added a background of 236 safe drugs—drugs with no known liver toxicity according to the information in the Spanish DILI Registry, SPC, and the negative DILI cases published by Chen et al. [25], Greene et al. [26] and Xu et al. [27]. Using information from the three aforementioned articles enabled us to compile a more extensive list of drugs with no known hepatotoxicity potential than what would have been possible if only using information from the Spanish DILI Registry. We filtered out herbal medications, dietary products, anabolic steroids, biological products and drug combinations; we also excluded drugs for which there is no clear toxicity information or discrepancies between the registries [28,29,30,31].

#### 2.1.1. Drug-Protein Association

We obtained dP interactions from ChEMBL v27 [32]. For each drug in our set, we retrieved all the human proteins annotated in UniProt/Swiss-Prot [33], i.e., proteins that have been characterised and manually curated. Therefore, these are highly reliable targets that bind to the drug with high affinity. We excluded low-confidence and low-affinity drug–protein interactions using the guidelines published by Hu and Bajorath [34], which we implemented in our previous work [17] (see Appendix A). ChEMBL is a database of bioactive drug-like small molecules and their bioactivities that offers curated data from the primary scientific literature, which allows us to obtain biologically meaningful drug–protein associations through the pChEMBL value, the negative logarithmic transformation of the affinity between the drug and its protein target, i.e., the lowest drug concentration at which the drug remains active against the target. For each drug, we kept the proteins reported to establish a direct binding interaction with the drug (assay_type = B; relationship_type = D; target_type = ‘SINGLE PROTEIN’). For drug–target interactions, we excluded non-specific interactions between drugs and biological targets by filtering out weak activities (i.e., the activity of a drug against a human protein target should be stronger than 10 μM, where activity includes IC50, EC50, XC50, AC50, Ki, Kd; pchembl_value ≥ 5). We considered protein targets for source assays of functional (F) or binding (B) types, activity measurements of exactly (standard_relation ‘=‘) the type Ki (inhibition constant) or IC50 (inhibition concentration) lower or equal to 10 μM standard activity units and without an activity comment of ‘Inactive’, ‘Not Active’ or ‘Inconclusive’.

#### 2.1.2. Drug-Domain Associations

We associated the drugs in our dataset with protein domains derived from Pfam—a database of curated protein domains based on a classification of homologous proteins in families. Protein domains are compact units of protein structure—they are the building blocks of proteins [35] that combine to form diverse proteins with different functions [36]. Ligand–domain interactions mediate Ligand–protein interactions; similarly, domain–domain interactions dominate protein–protein interactions [37]. Therefore, drug binding sites lie in Pfam domains [38], supporting the idea that the domain can be the druggable entity in a protein target [34]. We have inferred the associations between drugs and domains through the statistical overrepresentation of the protein targets of a drug among the relatives belonging to a domain (Pfam family), as shown in our previous work. In essence, for each drug, we look for the Pfam families that are significantly enriched in the targets of the drug based on the binomial test. Thus, for each possible drug–domain pair, we calculated a *p-value* (Benjamini–Hochberg corrected for multiple testing) that tells us if the protein targets of the drug tend to be members of the Pfam family that corresponds to the domain (see Appendix A). We refer the reader to our paper *Structural and functional view of polypharmacology* [17] for further details on the statistics of the drug–domain associations.

### 2.2. Integration of Drug–Protein and Drug–Domain Data. Drug Similarity Network

Figure 1 shows the protocol followed to integrate the protein and domain data into a drug similarity network. First, for each drug of our initial drug dataset, we have established their protein targets and Pfam domain targets in terms of interaction profiles; each drug was thus assigned a set of protein targets (the drug’s protein profile) and another set of domain targets (that constitute the drug’s domain profile). Then, for each pair of drugs (*i*,*j*), we measured the similarity between their protein profiles—and between their domain profile—by using the hypergeometric index as described by Bass et al. [39]. Each drug is represented by the set of its targets (i.e., the drug profile); the hypergeometric index is a similarity measure that reflects the proportion of overlap between the profiles of two drugs and considers only the number of shared elements between two drug profiles. The hypergeometric index is the minus log-transformed probability of finding a more significant profile overlap than that observed between drug *i* and drug *j* (see Figure 1A). In order to integrate the protein profiles similarity and the domain profiles similarity of each pair of drugs into one coherent dataset, we obtained a two-component vector for each drug pair. One component of the vector is the hypergeometric index that measures the similarity of the drug’s protein profiles. The other component is the hypergeometric index corresponding to the similarity of the drug’s domain profiles (Figure 1B). Therefore, the angle of such a vector (γ in Figure 1B) represents the relative importance of the protein profile similarity and the domain profile similarity in the drug–drug association.

In contrast, the module of the vector (α in Figure 1B) measures the intensity of the connection between the two drugs. When we extend this to all the possible drug pairs, we obtain a distribution of vector angles and a matrix—the drug similarity matrix—the coefficients of which are the modules of the vectors for each drug pair (α; see Figure 1C). Finally, the drug similarity matrix represents a network—the drug similarity network—in which drugs are linked if their targets (proteins and domains) are the same. Consequently, the values assigned to the interaction between each pair of drugs reflect their combined protein-based and domain-based similarity.

A cut-off value of 3 in the hypergeometric indices means that the probability of finding greater protein profiles overlap and domain profiles overlap between two drugs (i.e., a better drug–drug association) is lower than 0.001. Furthermore, the angles (γ in Figure 1B) are close to 45°, which means that protein profiles and domain profiles contribute to the interaction between drugs. In contrast, for more relaxed cut-offs of the hypergeometric index, the drug–drug interaction is described by domain data, whereas for more strict cut-offs, these drug associations are described predominantly by protein data (see Figure 2).

### 2.3. Topology and Community Analysis of the Drug Similarity Network

The drug similarity network (as many other networks) shows some topological features that describe its emergent properties, such as the presence of hubs and central drugs. Hubs are drugs with many more links than the average; central drugs are in privileged positions in the network structure. The drug similarity network can be assortative—if most of the links run between drugs of the same type, and exhibit a community structure, i.e., groups of drugs tightly interconnected that form communities or modules. We have analysed the topology of the drug similarity network with the iGraph R library [40].

The modular design of molecular networks captures the modularity of cellular function, i.e., cellular functions are carried out by modules made up of many species of interacting molecules [41]. Therefore, the groups of related elements in the network that form a community correspond to functional subunits. Thus, for example, in our drug network, a group of drugs that form a network module—named here a drug community—may affect the same biological processes and have similar mechanisms of action, for they share the same targets.

The drugs in our network can participate in multiple communities because these are overlapping rather than separate groups of drugs, which imposes the problem of determining each drug’s correct membership (or memberships). Because of this, we have identified the communities in our drug network based on the idea of the link communities [42] that depicts a community as a set of closely similar links instead of assuming a community to be a set of drugs tightly interconnected. Link communities group together shared interactions between drugs instead of grouping the drugs themselves to detect communities that correspond to similar molecular mechanisms while capturing the feature that drugs can belong to multiple communities. Since our drug communities comprise drugs with different degrees of liver toxicity, we have ranked them based on a DILI score that is calculated as ${\mathrm{DILI}}_{\mathrm{score}}=\sqrt{{\left(100\times \frac{\mathrm{td}+\mathrm{vtd}}{d}\right)}^{2}+d^{2}}$; where d, td and vtd are the number of drugs, the number of toxic drugs and the number of very toxic drugs in a drug community, respectively.

To further characterise the mechanisms by which the drugs in a community favour DILI, we have analysed the functional role in KEGG [43] and Gene Ontology [44] of the protein targets that define the links among drugs in each community.

We have made available all the code needed to reproduce our data in a code repository at https://github.com/fmjabato/dili_pipeline accessed on 19 May 2022.

## 3. Results and Discussion

### 3.1. Topology of the Drug Similarity Network

#### 3.1.1. Hubs and Drug-Target Data Completion

Many biological networks are scale-free [45]. The probability *P_k_* that a node links with *k* other nodes, i.e., the distribution of the number of neighbours of a node, follows a power law. Scale-freeness is easier to visualise in a logarithmic plot of *P_k_* vs. *k* as a line with a negative slope. The scale-freeness implies that the network organises around a few nodes with many neighbours (or a high degree), dubbed hubs, among many other network properties. In our drug similarity network, this would mean that a few drugs have targets (proteins and domains) that are also targets of many other drugs in the network; hence, a few drugs would link to many other drugs. The scale-freeness of drug networks reflects the incompleteness of the drug–target data from which the network comes—due to limited resources, drugs are only tested against a limited number of off-target proteins instead of being tested systematically through a large panel of protein targets for the sake of acquiring knowledge about their complete pharmacological profile. Mestres et al. [22] showed that drug networks modelled from limited and biased drug-target data tend to be scale-free with a degree distribution following a power law. As more drug–target data are added, the degree distribution of the network loses its power-law shape. The degree distribution of our drug network (202 drugs and 2695 links among them) does not fit a power law (Kolmogorov–Smirnov test *p-value* = 6 × 10^−10^). Figure 3 shows the degree distribution of our drug network and a scale-free network of the same size. We can see that our drug network deviates from the scale-free behaviour, which we interpret as a sign of the completeness and quality of the drug–target data we used in its modelling, according to Mestres et al. above. Furthermore, no drugs are hogging many connections, which means that there are no drugs with the same targets as many other drugs; in other words, there are no privileged targets widely spread throughout the system to which most of the drugs are connected.

#### 3.1.2. Assortative Mixing in the Drug Similarity Network

We wanted to assess the assortativity of the drug similarity network, that is, the ability of drugs to be preferentially connected to drugs of the same type, i.e., we wanted to know if DILI drugs connect with other DILI drugs more often than with safe drugs. To this end, we modelled two networks of the same size as our drug network, the nodes of which are annotated with two types so that they are in the opposite extremes of the assortativity spectrum. In the assortative network, nodes of one type seldom connect with nodes of the other type, whereas in the unassortative network, one type tends to connect with the other type (see Figure 4). The assortativity coefficient (*A*) is positive if the network is assortative, i.e., if DILI drugs are prone to connect with DILI drugs, and negative otherwise. We observe that although the drug similarity network is not entirely assortative, it leans towards assortative mixing (*A* = 0.06). Therefore, we can find pockets of DILI drugs linked preferentially with DILI drugs on a general background of drugs connected regardless of their type.

This result has two important implications: (i) Since drugs are linked because they have common targets, if DILI drugs tend to connect with DILI drugs, they have some targets in common. Nevertheless, there will be targets shared preferentially among safe drugs. Moreover, (ii) the drug similarity network is modular; we will find groups of drugs tightly connected, and these modules will be composed primarily of drugs of the same type. Based on these implications, we decided to analyse the modularity of the drug network further and characterise the drug communities in terms of their preferential targets and their safe/DILI drug composition.

### 3.2. Modularity and Community Structure of the Drug Similarity Network

The modularity of a network reflects the ability of the drugs to aggregate in drug communities or groups of drugs intensely connected. The modularity coefficient (*M*) measures how well the network can be divided into groups of interconnected drugs. As we did with the assortativity analysis above, to gauge the modularity of our drug network, we have modelled two networks at the opposite ends of the modularity range (Figure 5). We can see that our network is modular, and, taken together with the assortativity above, communities seem to incorporate DILI and safe drugs. Some of them are mainly composed of DILI drugs or safe drugs. Hence, we further characterised these communities through their DILI score and found several drug communities with high DILI scores (i.e., communities mainly composed of DILI drugs, including withdrawn drugs due to their toxicity) that share the same targets. Although we observe a trend in which DILI drugs gather together in the same communities, there is no absolute separation between DILI drugs and drugs that are considered safe. Except for the two communities with higher DILI scores in Table 1 containing toxic drugs, the drug communities identified in our drug network contain a mix of toxic and safe drugs.

Since any two drugs with common targets (proteins or Pfam domains) are linked together in our network, the Pfam family that is more prevalently shared among the drugs in a community can describe the community (Table 1; see Appendix A for the recollection of all drug communities). The description of a drug community by a Pfam family suggests the potential mechanism by which the drugs exert their liver toxicity. In the following sections, we discuss the communities eminently associated with DILI, i.e., those with a DILI score of 50% or more. We observe clear segregation of the communities of toxic drugs based on their size: Five small communities with less than five drugs and four large communities with more than ten drugs. These large communities have higher DILI scores because they contain more toxic drugs than safe drugs. These communities with higher DILI scores are analysed below.

#### 3.2.1. Animal-Haem Peroxidase and EGF-Like Drug Community

The same drug community was associated with two Pfam domains: Animal Haem peroxidase (PF03098) and EGF-Like (PF00008). The animal Haem peroxidase domain catalyses several immunological reactions. The EGF-Like domain is in the extracellular domain of membrane-bound proteins or secreted proteins and seems to play a crucial role in leukocyte movements toward inflammatory stimuli [46]. The 12 drugs in this community are hepatotoxic non-steroidal anti-inflammatory drugs (NSAIDs)—three of them, nimesulide, lumiracoxib and bromfenac, were withdrawn from the market due to acute liver failure.

Since NSAIDs form a community in our drug network, they have common targets among the animal haem peroxidase and EGF-like domain protein families—for instance, the well-known NSAIDs targets prostaglandin H synthase 1 and 2 (PTGS1 and PTGS2), also referred to as cyclooxygenase 1 and 2 (COX-1 and COX-2). Thus, these families might inform us of the specific mechanistic pathways by which NSAIDs cause DILI. Many homologous proteins in these families (myeloperoxidase, lactoperoxidase, thyroid peroxidase, prostaglandin H synthase) perform diverse biological roles such as innate immune defence, hormonal biosynthesis and prostaglandins synthesis [47,48,49]. Our results evidence the immune system as a general biological theme in how NSAIDs cause DILI, even though these particular mechanisms are difficult to detail. The participation of the immune system in DILI was established long ago, whether through the human leukocyte antigen [50] or other mediators [51,52], observations were made with a single or few drugs. Our results suggest that most NSAIDs might cause DILI by disrupting the immune-response pathways. The connection between drug–protein binding leading to immune-response alterations and liver injury is becoming apparent with the use of checkpoint inhibitor-based cancer treatments [53].

#### 3.2.2. Sodium-Neurotransmitter Symporter Drug Community

A large drug community (29 drugs) was associated with the sodium-neurotransmitter symporter domain (Pfam: PF00209), a family of neurotransmitter transporters that catalyse several neurotransmitters’ uptake of amino acids and other molecules by a Na^+^ symporter mechanism. Twenty out of the 29 drugs in this community cause liver injury—two of them, nomifensine and suloctidi, present a severe risk of liver reaction and have been withdrawn from the market in Spain.

There are three transporters from this superfamily expressed in the liver: SLC6A1, SCL6A12 and SCL6A13; all of them are involved in GABA transport, which protects the liver against injuries caused by toxic elements such as ethanol and D-galactosamine [54]. The drugs associated with this superfamily in our work could easily escape from hepatic compartments and interact with SLCA transporters, for they have a considerable potential to cross biological barriers. Therefore, they might cause DILI by binding to the GABA transporters in the liver, thus inducing a reduction of GABA transport to the intrahepatic biliary epithelium and subsequently, a restrain of hepatocytes proliferation leading to pseudobile ductile formation, bile flow blockade and liver injury. In fact, many drugs with hepatotoxicity potential, such as bosentan, pazopanib and ketoconazole, have been reported to inhibit transporter activity in vitro by binding to the bile salt export pump transporter and/or various multi-drug resistance-associated proteins, interactions assumed to be associated with DILI development [55]. Furthermore, transporters such as the carnitine acyl-carnitine carrier (SLC25) are believed to play an important role in the mitochondrial entry leading to oxidative stress in hepatotoxicity caused by valproic acid [56].

Moreover, the DILI drugs in this community interact with proteins involved in calcium signalling, the dysregulation of which is a hallmark of both acute and chronic liver diseases. Thus, they would cause a direct perturbation of mitochondrial calcium levels that could result in acute liver injury [54]. Another group of off-targets for these drugs participate in linoleic acid metabolism. As a result, these drugs could dysregulate the linoleic acid metabolism producing conjugated linoleic acid related to intracellular oxidative stress, thus inducing intrahepatic cholestasis, which is a common manifestation of DILI [50].

#### 3.2.3. Olfactory Receptors (ORs) Drug Community

The community with the highest number of drugs in our study was associated with olfactory receptors (also known as odorant receptors; Pfam: PF13853) since they are abundant in the cell membranes of olfactory receptors. However, during the last decade, it has been found that some ORs function in several non-olfactory systems related to new functions as chemical receptors, mediators of tissue growth and regeneration or cell–cell cooperation. The community has 36 drugs, of which 18 are potentially hepatotoxic, including two drugs (mepazine and pipemazine) withdrawn from the market because of severe hepatotoxic reactions. The therapeutics categories related to this drug community include treatments of the nervous system (the primary group with 20 drugs), followed by cardiovascular and respiratory treatments with six drugs each. Although many drugs in the olfactory receptors’ community are related to the respiratory or nervous system, none includes ORs as their therapeutical target. Therefore, these drugs might produce toxicity through their interaction with off-targets, illustrating the idea that the polypharmacological profile is a critical element in understanding its toxicity.

The olfactory receptors present in non-olfactory tissues (ectopic ORs) are involved in multiple processes—such as the initiation of the hypoxic ventilatory responses, the induction of glucose homeostasis in diabetes and the processes of tumour cell proliferation, apoptosis, metastasis and invasiveness, among others [57]. We found similar functional roles for the proteins associated with the drugs in this community. According to our functional analysis using KEGG and GO, these off-targets are involved in several cancer pathways and vasoconstriction and blood circulation, respectively. Although the function of ectopic ORs is mainly unknown, recent studies suggest they influence liver cell metabolism—by modulating the triglyceride metabolism [58]—and are overexpressed in liver tumours [59], where they mediate the reduction of hepatocellular carcinoma progression [60]. Therefore, even though the effect of ectopic ORs in drug-induced liver injury has not been observed yet, we believe their potential mechanism in DILI could be related to alterations in the regeneration of hepatocytes, or an imbalance between the production of lipids and their oxidation or transport, that affect the ratio of monounsaturated fatty acids and saturated fatty acids leading to apoptosis and liver damage. This potential mechanism agrees with the low levels of TBL, ALT and ALP observed in this community. Since damage production is not direct, it would be a slow and gradual process, so these enzymes could not reach high values before showing the first symptoms of liver damage.

We have postulated three potential mechanistic pathways associated with DILI based on the three largest drug communities. The Animal-haem peroxidase/EGF-like drug community composes drugs with well-known hepatotoxicity potential. Although the exact pathway is not yet elucidated, peroxidases, such as myeloperoxidase, are inducers of reactive metabolites that could modulate immune responses. Sodium-neurotransmitter symporter proteins have not been clearly linked to DILI, although their action could lead to cellular stress and mitochondrial damage, which are believed to occur in DILI development. Unlike the former two mechanistic pathways that have been associated with DILI development previously, the Olfactory receptor community offers a novel mechanistic hypothesis. The idea that this community may introduce mechanisms such as hepatocyte regeneration and lipid imbalance is based on the current literature.

#### 3.2.4. Drug Communities with a Low Number of Drugs

We find a series of small drug communities (i.e., communities with three to four drugs each) to devise their liver toxicity through the Pfam family they are associated with. The domains associated with these communities of a low number of drugs are the ABC transporter transmembrane domain, cytochrome P450, the ligand-binding domain of nuclear hormone receptor, multi-antimicrobial extrusion protein and eukaryotic-type carbonic anhydrase. Each of these protein families participates in biological processes that could constitute potential toxicity mechanisms for these drugs, explaining their implication in DILI.

### 3.3. Drug Communities and Demographic-Clinical Characteristics

We explored whether the DILI score of our drug communities was coupled with the therapeutic groups in DILI cases to assess if any demographic, biochemical or clinical characteristics—among those compiled in the Spanish DILI registry—were more prevalent in our drug communities. We compared each of the drug communities discussed above with a background of drugs that do not belong to any community (see Table 2).

The demographic variables in the Spanish DILI Registry were distributed similarly among our drug communities, i.e., sex distribution is approximately 50% and age medians range from 49 to 56 years (background median age is 58). However, we found community-specific patterns for some clinical and biochemical parameters. The latency and duration of treatment are significantly different from the background (62 vs. 22 days *p* < 0.0005 and 38 vs. 22 days *p* < 0.01, respectively) in the sodium-neurotransmitter symporter community. In this drug community, the presence of autoantibodies—such as antinuclear, antimitochondrial and anti-smooth muscle antibodies—was slightly lower than the background (autoantibodies present in 11% of the drug community and 24% of the background, *p* < 0.05). However, these values are below the typical prevalence of autoantibodies reported [45,46]. The damage profile of the sodium-neurotransmitter symporter community was eminently hepatocellular and significantly higher than the background (76% vs. 62%, *p* < 0.05), which we associate with a decrease in ALP values (1.2 vs. 1.6 ULN, *p* < 0.05) and a non-significant reduction in total bilirubin (2.14 vs. 5.0 ULN, *p* = 0.09).

The patients treated with drugs in the animal-haem peroxidase drug community show similar clinical and biochemical properties to the background, except eosinophilia presentation was less frequent in this group (16% vs. 24%, *p* < 0.05). The reason for this is unknown but could stem from variations in underlying non-hepatic conditions and consequently not be a true reflection of hepatic eosinophil state. Finally, the cases induced by drugs associated with the ORs community presented preferentially mild severity that contrasts with the background preferent moderate severity, though this effect is not statistically significant. 

### 3.4. Drug Communities and Drug Properties

Table 3 shows the physicochemical and pharmacokinetic variables of the drugs in the communities (sodium-neurotransmitter symporter, animal-haem peroxidase and ORs) compared with the drugs’ background not forming drug communities.

Even when the sodium-neurotransmitter symporter drugs do not have significant differences in their physicochemical properties, their pharmacokinetics deviates from the background: Higher rates of hepatic metabolism than the rest of the DILI population (100% vs. 65%, *p* < 0.005); longer half-life in the organism (11 vs. 6.5 h, *p* < 0.05); higher lipophilia values and lipoaffinity index (4.01 vs. 2.27, *p* < 0.001; 8.54 vs. 5.28, *p* < 0.0005, respectively). ORs drugs manifest a similar pharmacokinetic profile—a high-rate hepatic metabolism (95% vs. 65%, *p* < 0.005), longer half-life (14.1 h. vs. 6.5 h., *p* < 0.05) and higher lipophilia and lipoaffinity (3.9 vs. 2.27, *p* < 0.005; 8.6 vs. 5.28, *p* < 0.001, respectively). Consequently, these drugs have a high potential to cross biological barriers and remain disseminated across the organism for a long time. By contrast, the drugs associated with the animal-haem peroxidase domain do not have different pharmacokinetic profiles from the rest of DILI drugs but present different physicochemical properties. These drugs have a lower molecular weight (261 vs. 377, *p* < 0.05) and lower number of rings and heterorings (2.0 vs. 2.76, *p* < 0.05; 0.53 vs. 1.32, *p* = 0.01), which suggests that the animal-haem peroxidase drugs induce liver damage through particular mechanistic pathways, but do not inform us of their nature and details.

## 4. Conclusions

We present a systems pharmacology approach to tackle the complexities that DILI sets forth. We leverage the idea of drug polypharmacology, i.e., drugs often have multiple molecular targets, to correlate the liver toxicity caused by a drug with its molecular targets, both intended targets and off-targets, thus complementing and expanding other systems-level studies that base drug toxicity on drug structure, such as [61].

We have modelled a drug network where drugs are connected if they share targets, which is modular and assortative. That is, the drugs form communities where DILI drugs link preferentially with DILI drugs. This community structure of our drug network revealed several mechanisms by which drugs can damage the liver, such as disrupting pathways related to the immune system; promoting oxidative stress by the dysregulation of the linoleic acid metabolism; interfering with the liver GABA transport system or promoting apoptosis by altering the balance between monounsaturated and saturated fatty acids. Thus, even though we cannot pinpoint precise pathways or particular mechanisms of DILI, our systems-level analysis suggests general mechanisms or biological themes underlying DILI that merit further investigation.

## Figures and Tables

**Figure 1 genes-13-01292-f001:**
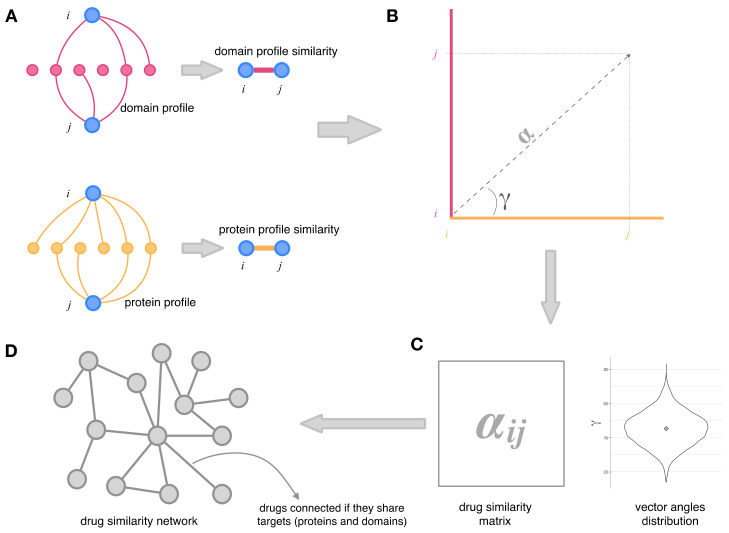
Integration of domain and protein data in a drug similarity network. (**A**) Drugs *i* and *j* are associated if they target the same domains or the same proteins; (**B**) hypergeometric indices measuring the similarity of the domains profiles between *i* and *j* (pink) and the similarity of the protein profiles (yellow) are represented as a vector; (**C**) drug similarity matrix obtained from the modules (α) of the hypergeometric indices and distribution of the angles of the vectors obtained for all drug pairs; (**D**) the drug similarity network is built from the drug similarity matrix.

**Figure 2 genes-13-01292-f002:**
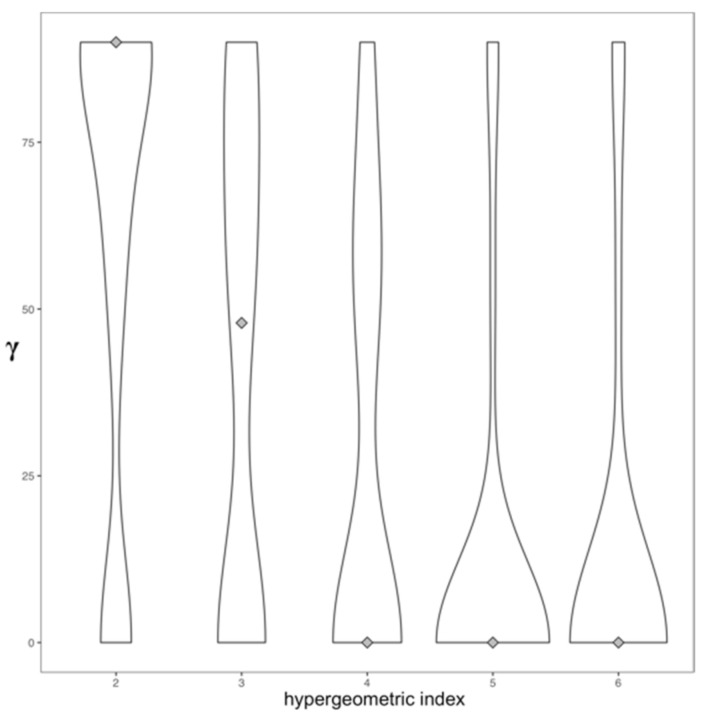
Distribution of drug similarity vectors. The angle (γ) of the vectors formed by the similarities of the drug profiles depends on the threshold used for the hypergeometric index.

**Figure 3 genes-13-01292-f003:**
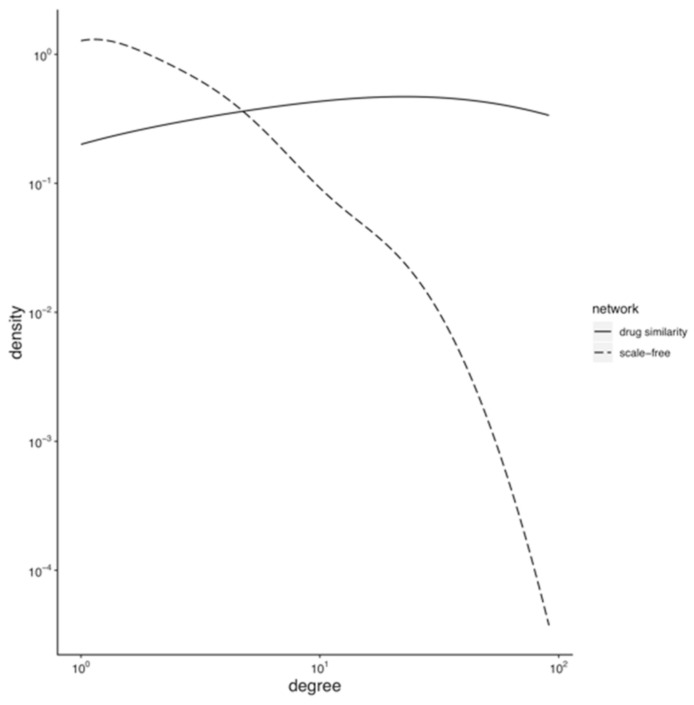
Degree distribution of the drug similarity network. The degree distribution of the drug similarity network (full line) is compared with a scale-free model of the same size (dashed line).

**Figure 4 genes-13-01292-f004:**
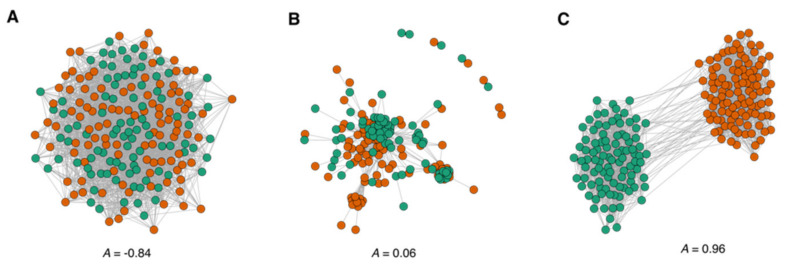
Assortative mixing of the drug similarity network. Nodes of different types are represented in green and orange circles (**A**) Assortative mixing of a network model in the low end of the assortativity range. (**B**) Assortative mixing of the drug similarity network. (**C**) Assortativity mixing of a network model in the upper end of the assortativity range.

**Figure 5 genes-13-01292-f005:**
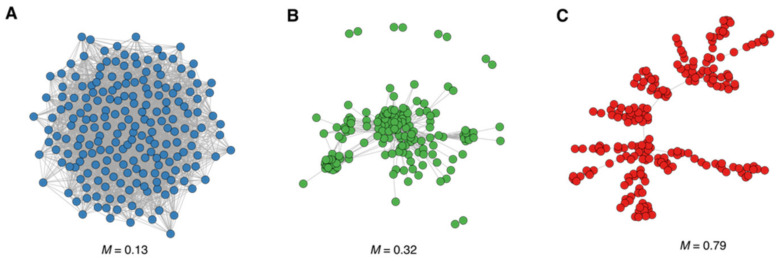
Modularity of the drug similarity network. The modularity (*M*) of the drug similarity network (**A**) is compared with a network model in the low end of the modularity range (**B**) and with a network model in the upper end of the modularity range (**C**).

**Table 1 genes-13-01292-t001:** Communities in the drug similarity network.

Domain Name	Pfam ID	Toxic Drugs, n	Non-Toxic Drugs, n	DILI Score	Therapeutics Categories (n Drugs)	
**EGF-like domain**	PF00008	12	0	101	NSAIDs	(12)
**Animal haem peroxidase**	PF03098	12	0	101	NSAIDs	(12)
**Sodium-neurotransmitter symporter domain**	PF00209	20	9	75	Nervous system	(12)
					Cardiac system	(8)
					Respiratory	(2)
					Antineoplastic	(2)
					Gynecological	(2)
					Alimentary	(1)
					Blood formation	(1)
					Dermatological	(1)
**Ligand-binding domain of nuclear hormone receptor**	PF00104	2	1	67	Sexual hormone	(2)
					Cardiac system	(1)
**ABC transporter transmembrane region**	PF00664	2	1	67	Nervous system	(1)
					Cardiac system	(1)
					Antineoplastic	(1)
**Olfactory receptor**	PF13853	18	18	62	Nervous system	(20)
					Cardiac system	(6)
					Respiratory	(6)
					Alimentary	(2)
					Gynaecological	(1)
					Dermatological	(1)
**Eukaryotic-type carbonic anhydrase**	PF00194	2	2	50	Cardiac system	(2)
					Antineoplastic	(1)
					Nervous system	(1)
**Multi-antimicrobial extrusion protein**	PF01554	2	2	50	Anti-infective	(2)
					Antineoplastic	(1)
					Respiratory	(1)
**Cytochrome P450**	PF00067	2	2	50	Dermatological	(3)
					Various	(1)
**7 transmembrane receptor (rhodopsin family)**	PF00001	19	25	62	Nervous system	(18)
					Respiratory	(8)
					Alimentary	(5)
					Gynaecological	(5)
					Cardiac system	(4)
					Dermatological	(2)
					Antineoplastic	(1)
					Hormonal	(1)
**Sugar (and other) transporter**	PF00083	1	2	33	Alimentary	(1)
					Cardiac system	(1)
					Anti-infective	(1)
**Major Facilitator Superfamily**	PF07690	1	2	33	Alimentary	(1)
					Cardiac system	(1)
					Anti-infective	(1)
**FAD-binding domain**	PF00890	1	2	33	Nervous system	(2)
					Cardiac system	(1)
**NAD(P)-binding Rossmann-like domain**	PF13450	1	2	33	Nervous system	(2)
					Cardiac system	(1)
**DEAD/DEAH box helicase**	PF00270	1	4	21	Nervous system	(2)
					Cardiac system	(1)
					Respiratory	(1)
					Antineoplastic	(1)

**Table 2 genes-13-01292-t002:** Drug communities and therapeutic groups of DILI cases.

	PF00209	*p-Value **	PF03098	*p-Value* *	PF13853	*p-Value* *	Remaining DILI Cases
	(n = 45)		(n = 141)		(n = 40)		(n = 757)
**Demographics**							
Age, median (IQR)	49 (41–68)	0.6285	55 (40–69)	0.3496	56 (43–69)	0.9831	58 (42–69)
Women, n (%)	24 (53)	0.4828	71 (50)	0.6002	20 (50)	0.8006	363 (48)
**Clinical parameters**							
Duration of treatment, median days (IQR)	62 (29–150)	0.0001	23 (8–58)	0.4333	37 (15–77)	0.0559	22 (8–63)
Time to onset, median days (IQR)	38 (19–122)	0.0092	25 (7–61)	0.3342	24 (8–78)	0.5889	22 (9–60)
**Severity, n (%)**		0.1921		0.1254		0.0165	
Mild	20 (45)		43 (32)		21 (55)		230 (31)
Moderate	19 (43)		79 (59)		13 (34)		429 (58)
Severe	4 (9)		3 (2)		3 (8)		52 (7.0)
Fatal/Transplant	1 (2)		8 (6)		1 (2)		27 (3.6)
Eosinophilia, n (%)	7 (17)	0.3127	21 (16)	0.0416	8 (28)	0.5376	178 (24)
Lymphopenia, n (%)	9 (26)	0.6801	23 (20)	0.5260	6 (16)	0.2495	146 (23)
Positive autoantibody titres ^§^, n (%)	4 (11)	0.0394	29 (25)	0.9494	5 (16)	0.2617	148 (24)
**Type of liver injury, n (%)**		0.0413		0.8202		0.3628	
Hepatocellular	34 (76)		84 (61)		27 (67)		448 (62)
Cholestastic & Mixed	11 (24)		53 (38)		13 (32)		273 (30)
**Liver profile values at onset xULN, median (IQR)**							
TBL	2.14 (0.7–6.2)	0.0900	4.2 (1.2–9.3)	0.3855	1.5 (0.6–4.4)	0.0099	5 (1.15–10)
AST	5.6 (2.1–15)	0.5378	8.4 (2.7–26)	0.2773	5.0 (2.4–13)	0.2405	6.4 (3.0–19)
ALT	7.3 (4.0–28)	0.5619	9.5 (4.5–30)	0.2847	8.1 (4.3–16)	0.0582	9.8 (4.7–24)
GGT	4.0 (2.1–10)	0.5000	6.6 (2.8–20)	0.1910	5.9 (2.2–9.4)	0.3671	5.4 (2.7–10)
ALP	1.2 (0.7–4.7)	0.0420	1.7 (1.1–2.9)	0.0818	1.0 (0.7–2.0)	0.0316	1.6 (1.0–4.2)

Abbreviations: ALP, alkaline phosphatase; ALT, alanine transaminase; AST, aspartate transaminase; d, days; DILI, drug-induced liver injury; GGT, γ-glutamyltransferase; IQR, interquartile range; TBL, total bilirubin; xULN, times upper limit of normal. * *p-values* provided for the comparison with the “remaining DILI cases” group. ^§^ anti-nuclear, anti-smooth muscle, anti-mitochondrial and/or liver kidney microsomal type 1 antibodies

**Table 3 genes-13-01292-t003:** Physicochemical, pharmacokinetic and pharmacodynamics properties of drugs based on their classification (community or therapeutic group).

	PF00209	*p-Value **	PF03098	*p-Value **	PF13853	*p-Value **	Remaining DILI Drugs
	(n = 20)		(n = 15)		(n = 22)		(n = 141)
**Physicochemical**							
Molecular weight, mean	331	0.3577	261	0.0124	364	0.7271	377
Number of rings, mean	3.09	0.3118	2.0	0.0213	3.63	0.0104	2.76
Heterorings, mean	0.90	0.1069	0.53	0.0089	1.63	0.2390	1.32
**Pharmacokinetics**							
Half-life, median h	11	0.0234	2.2	0.2913	14.1	0.0130	6.5
Lipophilicity (LogP), median (range)	4.01	0.0008	2.87	0.4447	3.9	0.001	2.27
	(0.51–6.3)		(−0.9–4.3)				(−6.8–8.3)
Plasma protein binding (%), median (range)	95	0.0204	99	0.0001	93	0.0099	84
	(27–99.3)		(55–99.5)		(56–99.98)		(1–99.9)
Hepatic metabolism	20 (100)	0.0039	13 (87)	0.1630	20 (95)	0.0134	98 (65)
≥50%, n (%)							
Enterohepatic circulation, n (%)	8 (42)	0.1908	2 (13)	0.2345	10 (47)	0.0618	38 (27)
Reactive metabolite formation, n (%)	8 (40)	0.7871	7 (46)	0.4574	9 (41)	0.7164	52 (37)
Mitochondrial liability, n (%)	13 (65)	0.2198	13 (87)	0.0073	13 (59)	0.4457	71 (50)
Lipoaffinity, mean	8.54	0.0003	4.68	0.6014	8.6	0.0006	5.28
**BDDCS**		0.0090		0.5158		0.0048	
Class 1 (↑solub/↑hep met)	12 (60)		4 (57)		14 (64)		39 (28)
Class 2 (↓solub/↑hep met)	8 (40)		8 (29)		8 (36)		60 (43)
Class 3 (↑solub/↓hep met)	-		2 (14)		-		25 (18)
Class 4 (↓solub/↓ hep met)							16 (11)

Abbreviations: BDDCS, Biopharmaceutical Drug Disposition Classification System; h, hours; ↑solub/↑hep met, high solubility/extensive hepatic metabolism; ↓solub/↑hep met, low solubility/extensive hepatic metabolism; ↑solub/↓hep met, high solubility/poor hepatic metabolism; ↓solub/↓ hep met, low solubility/poor hepatic metabolism. * *p-values* provided for the comparison with the “remaining DILI drugs” group.

## Data Availability

Clinical data analysed in this study were obtained from the Spanish DILI Registry database, a privately-owned database maintained by the registry’s coordinating centre in Malaga, Spain. The database is not publicly available. The data generated in this study is available as Appendix A. The software developed for this analysis is available on GitHub: https://github.com/fmjabato/dili_pipeline, accessed on 19 May 2022.

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
