# Peer review of "Identification of New Toxicity Mechanisms in Drug-Induced Liver Injury through Systems Pharmacology"

_genes, 2022, doi:10.3390/genes13071292_

Round 1
Reviewer 1 Report
DILI remains a serious issue in terms of both patient health and drug development, and lack of mechanistic insights in DILI continues to hamper progress in this field. This manuscript exploits a systems pharmacology approach to gain understanding of novel DILI mechanisms.
The authors explain the development of a drug network based on shared targets (proteins or domains) between drugs. The collected information is compiled in a so-called drug similarity network, which is then analysed according to topologies and communities.
Based on their analyses, the authors propose several mechanisms by which drugs appear to damage the liver, providing directions for further investigations.
The methodological approach is interesting and generally well described although some further clarifications/revisions are recommended (see below). Overall, the authors should try to bring more structure to their manuscript. Also, the authors can improve their manuscript in terms of comparison to other studies in this field.
The specific comments are thus as follows:
1. It will be impossible for the readers to reproduce the systems pharmacology approach as developed by the authors. Full scripts and complete tables with findings should be made available to the reader. As an example, on Ln 334, the authors refer to a community of 29 drugs, but the full list of these compounds does not seem to be available.
2. The authors should make it clearer as to which findings (in terms of involved pathways in DILI) are novel versus have been identified based on previous research.
3. The authors classify their approach as a systems pharmacology approach, which implies a bottom-up approach. However, their methodology is rather a statistical, top-down approach. The introduction would benefit from a better perspective on various approaches for predicting DILI, and the difference of the approach used in the present study with other studies in this field. In particular, in the second paragraph, no reference to existing systems pharmacology/toxicology approaches (e.g. https://www.ncbi.nlm.nih.gov/pmc/articles/PMC5359062/).
4. From a methodological perspective, the manuscript would benefit from a schematic workflow, i.e. what is now partially represented already in Figure 1. In addition, a text box with study-specific definitions, such as for ‘drug similarity network’, ‘hypergeometric index’, ‘drug communities’, ‘assortative network’, ‘modularity’,…
5. Ln. 102-113: the authors do not really justify the selection of data sources for compiling drug, protein and domain data. For instance, why dugs from the Spanish DILI registry (alone), and why the publications by Chen, Greene and Xu?
6. Ln. 115/116: are these dP interaction data based on in vitro, in vivo and/or in silico studies? The authors should probably add a short description here.
7. Ln. 158: the manuscript would benefit from a bit more elaborate description on the hypergeometric index as described by Bass et al. The method described by Bass et al. refers to gene similarity. To what extent can this method be applied to proteins without modifications?
8. Ln. 183-185: this needs rephrasing as it presently does not read sufficiently clear and is actually inaccurate.
9. Table 1 legend is missing
10. Ln 321-329: this section is lacking citations to relevant literature. The targets should be mentioned in full at least once.
11. Table 3: replace “lipophilia” with “lipophilicity”
12. Table 3: the row for plasma protein binding is empty (shifted?)
Author Response
Ms Mia Gao
Assistant Editor
Genes
Subject: Submission of revised manuscript genes-1757650
Dear Ms Gao:
Thank you for your email dated June 14th 2022, enclosing Reviewer #1 comments. We appreciate them and their suggestions which we believe have strengthened our manuscript. We have edited our manuscript for clarity and addressed the reviewer's concerns. Below we describe how we have revised our paper.
Reviewer #1: It will be impossible for the readers to reproduce the systems pharmacology approach as developed by the authors. Full scripts and complete tables with findings should be made available to the reader. As an example, on Ln 334, the authors refer to a community of 29 drugs, but the full list of these compounds does not seem to be available.
We appreciate this observation. We agree that reproducibility is a significant issue we have not appropriately addressed in our work. We have made all the code and scripts available in a GitHub repository and complete tables with findings in supplementary materials. The code repository can be accessed at https://github.com/fmjabato/dili_pipeline and contains all the code necessary for replicating our work, such as querying the databases, modelling the drug-target associations and the drug network, and obtaining the drug communities. We have included a mention of the GitHub repository in the manuscript (lines 286-287).
Reviewer #1: The authors should make it clearer as to which findings (in terms of involved pathways in DILI) are novel versus have been identified based on previous research.
We agree with Reviewer #1 in that novel, and corroborative findings could be highlighted more clearly. The idiosyncratic nature of DILI makes it a challenge both in clinical practise as well as on a molecular level. Various theories on the underlying mechanism of DILI have been reported to date. Some are supported by experimental data, analytical data and, to some degree, bibliography. However, none of these theories is conclusive in identifying mechanistic pathways involved in DILI development. In the current study, we identified three large drug communities, of which two corroborate earlier findings and one associated with olfactory receptors, introducing a new concept into the area of the DILI mechanism. To emphasise this, we added the following text to the manuscript (lines 485 to 495): "We have postulated three potential mechanistic pathways associated with DILI based on the three largest drug communities. The Animal-haem peroxidase/EGF-like drug community composes drugs with well-known hepatotoxicity potential. Although the exact pathway is not yet elucidated, peroxidases, such as myeloperoxidase, are inducers of reactive metabolites that could modulate immune responses. Sodium-neurotransmitter symporter proteins have not been clearly linked to DILI, although their action could lead to cellular stress and mitochondrial damage, which are believed to occur in DILI development. Unlike the former two mechanistic pathways that have been associated with DILI development previously, the Olfactory receptor community offers a novel mechanistic hypothesis. The idea that this community may introduce mechanisms like hepatocyte regeneration and lipid imbalance are based on the current literature."
Reviewer #1: The authors classify their approach as a systems pharmacology approach, which implies a bottom-up approach. However, their methodology is rather a statistical, top-down approach. The introduction would benefit from a better perspective on various approaches for predicting DILI, and the difference of the approach used in the present study with other studies in this field. In particular, in the second paragraph, no reference to existing systems pharmacology/toxicology approaches (e.g. https://www.ncbi.nlm.nih.gov/pmc/articles/PMC5359062/)
We would like to draw Reviewer's #1 attention to the fact that systems pharmacology (as well as systems biology) can follow both bottom-up and top-down modelling philosophies. The modelling philosophy exposed in the publication they are suggesting is bottom-up, while ours is top-down. We believe we have appropriately described the approaches to studying DILI. However, we have included the Reviewer's #1 suggestion to enrich the introduction (lines 44 to 63).
Reviewer #1: From a methodological perspective, the manuscript would benefit from a schematic workflow, i.e. what is now partially represented already in Figure 1. In addition, a text box with study-specific definitions, such as for 'drug similarity network', 'hypergeometric index', 'drug communities', 'assortative network', 'modularity'
Reviewer #1: Ln. 158: the manuscript would benefit from a bit more elaborate description on the hypergeometric index as described by Bass et al. The method described by Bass et al. refers to gene similarity. To what extent can this method be applied to proteins without modifications?
We believe we have made a substantial effort to explain all the systems biology terms we use in our study, so a non-expert reader can understand what we do. All the terms picked out by Reviewer #1 as examples are extensively explained: drug communities in lines to; assortativity in lines 254 to 257 and lines 324 to 336; modularity and drug communities in lines 258 to 264 and lines 354 to 368. The hypergeometric index is not a metric specific to gene similarity. We have expanded its explanation (see lines 207 to 212), which clarifies how we measure drug similarity.
Regarding Figure 1, we believe it serves the purpose of a workflow since it explains all the steps we perform to obtain the drug similarity network from the drug-protein and drug-domain data. Figure 1 legend and the text accompanying Figure 1 clarify all the steps we follow, thus providing an extensive explanation of how we obtain the drug similarity network.
Reviewer #1: Ln. 102-113: the authors do not really justify the selection of data sources for compiling drug, protein and domain data. For instance, why dugs from the Spanish DILI registry (alone), and why the publications by Chen, Greene and Xu?
We thank Reviewer #1 for their comment. The Spanish DILI Registry is one of the largest DILI registries worldwide, with a large selection of causative agents. This makes it a valuable source of drugs with different degrees of hepatotoxicity potential. The additional works by Chen, Greene and Xu were included to verify the information from the Spanish DILI Registry as stated in the manuscript ("...we also excluded drugs for which there is no clear toxicity information or discrepancies between the registries..."). Furthermore, while the Spanish DILI Registry is a good source of DILI causative agents, its potential is limited when it comes to drugs without known hepatotoxicity potential. Hence, we used information from the three aforementioned studies to extend the list of drugs with no known hepatotoxicity. This information has been added to the manuscript (see lines 141 to 144): "Using information from the three aforementioned articles enabled us to compile a more extensive list of drugs with no known hepatotoxicity potential than what would have been possible if only using information from the Spanish DILI Registry."
Reviewer #1: Ln. 115/116: are these dP interaction data based on in vitro, in vivo and/or in silico studies? The authors should probably add a short description here.
We obtain the drug-protein data from ChEMBL, a very well-established database of bioactive drug-like small molecules and their bioactivities (e.g. binding constants, pharmacology and ADMET data). ChEMBL compiles data from different sources and presents the interaction between small molecules and targets based on their affinity, which is the relevant issue for our work. We have added a short description of the drug- protein data in ChEMBL (lines 154-156).
Reviewer #1: Ln. 183-185: this needs rephrasing as it presently does not read sufficiently clear and is actually inaccurate.
This is now corrected (see lines 241-243). We apologise for this typo.
Reviewer #1: Table 1 legend is missing
We have added the legend to Table 1. We discovered minor mistakes in Table 1 that have been corrected as well.
Reviewer #1: Ln 321-329: this section is lacking citations to relevant literature. The targets should be mentioned in full at least once.
We apologise for not including the full name of these targets. This has now been amended, and the text now reads: "...the well-known NSAIDs targets prostaglandin H synthase 1 and 2 (PTGS1 and PTGS2), also referred to as cyclooxygenase 1 and 2 (COX-1 and COX-2)." (see lines 403-404). In addition, we have included three new references to support current information on the biological roles of the indicated protein families.
Reviewer #1: Table 3: replace "lipophilia" with "lipophilicity". Table 3: the row for plasma protein binding is empty (shifted?)
We have corrected Table 3.
The reviewer's comments were constructive in highlighting the lack of clarity in our manuscript. Following their suggestions, we have substantially edited our manuscript and provided the code and supplementary materials. We believe we have met the Editor and the reviewer's concerns, and we hope our revised paper is now acceptable for publication in Genes.
On behalf of the authors, Aurelio A. Moya-Garcia

Reviewer 2 Report
The manuscript "Identification of new toxicity mechanisms in drug-induced liver injury through systems pharmacology" is an attempt to leverage a large data set of DILI-causing drugs and their interactions with a range of biologically relevant proteins into a discussion of potentially new mechanisms for DILI. The research methods show promise for the discovery of new targets; though the general mechanisms identified by the research in this paper have largely been previously identified, the confirmation of these mechanisms by the researchers' methods serves as some degree of validation. Generally the research is sound, but there are several ways in which this could be improved.
General comments:
Extra validation could be achieved by demonstrating the association of known DILI drugs with proteins with which they have been known to interact. For example, substantial work has been done associating DILI and non-DILI drugs with BSEP; demonstrating that the DILI drugs shown to inhibit BSEP in those papers are included in the ABC transporter node in this network would lend added plausibility to the research here. This is just an example; there are other in vitro experiments with other proteins that could be referenced here.
While potency of interaction is mentioned here, the dose of the compound and its likely concentration near the target is not; was the concentration of the drug considered in the decisions to include/not include certain interactions? If not, it may strengthen the research here to consider Cmax/Ki ratios (or whatever interaction coefficient is appropriate) rather than simply using the somewhat arbitrary cutoff of 10 uM.
Specific comments:
More information on the association constant would be helpful; the authors neglect to report the overall A value for their network and the description of how it is calculated could use more specifics, perhaps in a supplement.
Animal-haem peroxidase is described as an immune system modulator, and the authors suggest that the animal-haem peroxidase association with certain NSAIDs may describe a novel mechanism for immune system activation and hence DILI. However, the clinical analysis of these drugs showed that eosinophilia is decreased in the drugs associated with this particular Pfam category. This would seem to contradict the suggestion that immune activation is involved in this interaction's potential mechanism of toxicity. Can the authors comment on this seeming paradox?
Table 2 and Table 3 are difficult to read; a re-formatting is in order.
It is unclear what the supplementary materials have to do with the manuscript.
Author Response
Ms Mia Gao
Assistant Editor
Genes
Subject: Submission of revised manuscript genes-1757650
Dear Ms Gao:
Thank you for your email dated June 14th 2022, enclosing Reviewer #2 comments. We appreciate them and their suggestions which we believe have strengthened our manuscript. We have edited our manuscript for clarity and addressed the reviewer's concerns. Below we describe how we have revised our paper.
Reviewer #2: Extra validation could be achieved by demonstrating the association of known DILI drugs with proteins with which they have been known to interact. For example, substantial work has been done associating DILI and non-DILI drugs with BSEP; demonstrating that the DILI drugs shown to inhibit BSEP in those papers are included in the ABC transporter node in this network would lend added plausibility to the research here. This is just an example; there are other in vitro experiments with other proteins that could be referenced here.
We thank Reviewer #2 for this suggestion. The following sentences with associated references have now been added to the manuscript (lines 414-416): "The connection between drug-protein binding leading to immune-response alterations and liver injury is becoming apparent with the use of checkpoint inhibitor- based cancer treatments," and lines 432-438: "In fact, many drugs with hepatotoxicity potential, such as bosentan, pazopanib and ketoconazole, have been reported to inhibit transporter activity in vitro by binding to the bile salt export pump transporter and/or various multi-drug resistance-associated proteins, interactions assumed to be associated with DILI development [55]. Furthermore, transporters such as the carnitine acyl-carnitine carrier (SLC25) is believed to play an important role in mitochondrial entry leading to oxidative stress in hepatotoxicity caused by valproic acid [56]."
Reviewer #2: While potency of interaction is mentioned here, the dose of the compound and its likely concentration near the target is not; was the concentration of the drug considered in the decisions to include/not include certain interactions? If not, it may strengthen the research here to consider Cmax/Ki ratios (or whatever interaction coefficient is appropriate) rather than simply using the somewhat arbitrary cutoff of 10 uM.
We believe we do not use the term "potency" referring to the interaction between drugs and protein targets. We select drug-protein interactions based on affinity. We exclude low-affinity interactions between drugs and proteins so we can retrieve compounds that are selective to one target over other targets. ChEMBL implements a metric (pChEMBL) that integrates several comparable measures of half-maximal response concentration and affinity (IC50, XC50, EC50, AC50, Ki, Kd). Therefore, the threshold imposed on pChEMBL is a convenient (and widely used) way to filter out low-affinity drug-protein associations. Since many ways to measure affinity or the activity of a drug towards its targets are through the concentration of the drug, we illustrate in terms of concentration the chosen pChEMBL threshold.
Reviewer #2: More information on the association constant would be helpful; the authors neglect to report the overall A value for their network and the description of how it is calculated could use more specifics, perhaps in a supplement.
We do not know what association constant Reviewer #2 refers to. "Association constant" does not appear in our manuscript. Regarding the overall A value, the assortativity of the drug similarity network is shown in Figure 4. To clarify this issue, we show the assortativity coefficient for our network in the text (line 334). The definition of assortativity and further explanations of the concept are included in the manuscript (lines 254-257 and 324-336.
Reviewer #2: Animal-haem peroxidase is described as an immune system modulator, and the authors suggest that the animal-haem peroxidase association with certain NSAIDs may describe a novel mechanism for immune system activation and hence DILI. However, the clinical analysis of these drugs showed that eosinophilia is decreased in the drugs associated with this particular Pfam
category. This would seem to contradict the suggestion that immune activation is involved in this interaction's potential mechanism of toxicity. Can the authors comment on this seeming paradox?
Eosinophilia in Table 2 refers to peripheral blood eosinophilia and was determined as the number of eosinophils being >5% of total leukocyte count (please see Kuang FL. Approach to the patient with eosinophilia. Med Clin North Am 2020;104:1-14). Peripheral blood eosinophilia does not necessarily reflect tissue eosinophilia, in this case eosinophil state in the liver. Furthermore, the percentage of peripheral eosinophils can be misleading as a normal eosinophil count may appear as eosinophilia in a situation of decreased number of total leukocytes. Alternatively, eosinophilia may be disguised in a situation of increased leukocyte level. The protein family containing an animal haem peroxidase domain comprises proteins associated with various immune cell types, including neutrophils and monocytes. Hence, the effect of the drug community may not be uniform but affect various types of immune cells.
A different aspect to take into consideration is that many of the DILI cases featured in Table 2 also had additional underlying non-hepatic conditions, which can affect peripheral eosinophil count. For example, allergies, asthma, parasitic infections and cancer are commonly associated with increased peripheral eosinophil levels, while bacterial infections are more commonly caused by decreased eosinophil levels.
To clarify the issue for the reader, the following sentence has been included in the manuscript (lines 534- 536): "The reason for this is unknown, but could stem from variations in underlying non-hepatic conditions and consequently not be a true reflection of hepatic eosinophil state."
Reviewer #2: Table 2 and Table 3 are difficult to read; a re-formatting is in order.
We have formatted tables 2 and 3. We believe the information is more clear now. We thank #Reviewer 2 for their suggestion.
Reviewer #2: It is unclear what the supplementary materials have to do with the manuscript.
We apologise for this major mistake. Something went wrong during the submission process, and an example dataset was uploaded as the supplementary materials. This is now fixed with the correct supplementary material.
The reviewer's comments were constructive in highlighting the lack of clarity in our manuscript. Following their suggestions, we have substantially edited our manuscript. We believe we have met the Editor and the reviewer's concerns, and we hope our revised paper is now acceptable for publication in Genes.
On behalf of the authors, Aurelio A. Moya-Garcia
